# Genome-Wide SNPs Detect Hybridisation of Marsupial Gliders (*Petaurus breviceps breviceps* × *Petaurus norfolcensis*) in the Wild

**DOI:** 10.3390/genes12091327

**Published:** 2021-08-27

**Authors:** Monica L. Knipler, Mark Dowton, Katarina Maryann Mikac

**Affiliations:** 1Faculty of Science, Medicine and Health, School of Earth, Atmospheric and Life Sciences, University of Wollongong, Wollongong, NSW 2522, Australia; mk972@uowmail.edu.au; 2Faculty of Science, Medicine and Health, The School of Chemistry and Molecular Bioscience, University of Wollongong, Wollongong, NSW 2522, Australia; mdowton@uow.edu.au

**Keywords:** DArTseq, hybridisation, Petaurus, SNPs, genomics, conservation, habitat fragmentation

## Abstract

*Petaurus breviceps* and *Petaurus norfolcensis* have produced hybrids in captivity, however there are no reported cases of Petaurus hybridisation in the wild. This study uses morphological data, mitochondrial DNA, and nuclear genome-wide SNP markers to confirm *P. breviceps breviceps × P. norfolcensis* hybridisation within their natural range on the central coast of New South Wales, Australia. Morphological data identified a potential hybrid that was confirmed with next-generation sequencing technology and 10,111 genome-wide SNPs. Both STRUCTURE and NewHybrid analyses identified the hybrid as a *P. norfolcensis* backcross, which suggests an initial F1 hybrid was fertile. The mitochondrial DNA matched that of a *P. b. breviceps*, indicating that a *P. b. breviceps* female initially mated with a *P. norfolcensis* male to produce a fertile female offspring. Our study is an important example of how genome-wide SNPs can be used to identify hybrids where the distribution of congeners overlaps. Hybridisation between congeners is likely to become more frequent as climate changes and habitats fragment, resulting in increased interactions and competition for resources and mates.

## 1. Introduction

There are documented cases of Australian mammal hybridisation in captivity where parental pairings are known, but there is limited evidence of it occurring in the wild [1,2,3]. Mammal hybridisation was historically thought to be a rare occurrence in the wild, however, this perspective has started to shift as advances in next-generation sequencing technology provide fresh insights through genomics [4,5,6]. Genome-wide single nucleotide polymorphisms (SNPs) have successfully detected hybrids in wild populations of canids [7], and has also been used to detect hybridisation in other vertebrates such as Australian frogs [8]. Identifying hybrid zones is important for wildlife conservation, as hybrids are typically less fit than their parents due to outbreeding of genes from two species [9,10]. Additionally, not all hybrids reach maturity and those that do can be infertile [11]. This can be detrimental in small populations where infertile hybrids can take away resources and mates from a threatened species.

Habitat destruction and fragmentation are known to increase species interactions, which can, in turn, increase the chances of hybridisation [12]. Australia has experienced extremely high levels of deforestation, and species that share distributions are encountering each other more frequently in fragmented patches, leading to increased competition for resources [13]. The genus Petaurus consists of small, marsupial gliders that are particularly susceptible to habitat loss due to their arboreal nature [14,15]. Two Petaurus species that share the same distribution are *P. norfolcensis* and *P. breviceps*. Both species occur along the east coast of Australia, with *P. norfolcensis* sharing habitats in Queensland with *P. b. longicaudus* and *P. b. breviceps*, and the rest of its range with *P. b. breviceps* (Figure 1) [15,16]. *P. norfolcensis* is currently listed as a threatened species in South Australia, Victoria, and New South Wales, and common in Queensland (National Parks and Wildlife Act 1972, Flora and Fauna Guarantee Act 1988, Threatened Species Conservation Act 1995), while *P. breviceps* is listed as common across each state. The conservation status and listing of *P. breviceps* may change with the recent confirmation of the three species, where only *P. breviceps* was once recognised [15].

*P. breviceps* and *P. norfolcensis* are believed to have diverged into separate species at least 4.46 million years ago based on fossil evidence by Malekian et al. [17]. They are both morphologically similar to each other with a few distinguishing features. Notably, *P. norfolcensis* weighs significantly more than *P. breviceps* (190–330 g, 90–150 g, respectively) [18,19,20]. *P. norfolcensis* have thick, long tails that are exclusively black while *P. breviceps* can have a white-tip on their tail. The head length is longer in *P. norfolcensis* than *P. breviceps*, as *P. breviceps* has a round head and snub nose [18,20].

Despite the overlap in distribution across Australia, there are currently no known cases of *P. breviceps* and *P. norfolcensis* hybridisation in the wild. Interbreeding of *P. breviceps* and *P. norfolcensis* is possible given the two records of hybridisation in captivity reported by Fleay [2] and Zuckerman [3]. Smith [21] attempted to replicate this by interbreeding *P. breviceps* and *P. norfolcensis* in captivity in 1964, but all attempts were unsuccessful. Reproduction in *P. breviceps* and *P. norfolcensis* occurs year round with no separation of breeding seasons [22], increasing the potential of congener mating. Additionally, both species feed on similar sap, pollen, nectar, and invertebrate species [20,23,24]. This means that *P. breviceps* and *P. norfolcensis* can seek to occupy the same habitat fragment, especially if habitat fragmentation imposes pressure on interspecies competition.

We examined a region of eastern Australia where *P. b. breviceps* and *P. norfolcensis* are known to coexist in the wild. Morphological data, mitochondrial DNA (mtDNA), and nuclear data in the form of genome-wide SNPs are used to identify pure *P. b. breviceps*, pure *P. norfolcensis*, and potential hybrids.

## 2. Materials and Methods

### 2.1. Genetic Sampling and Morphological Measurements

This study used genomic DNA collected from 21 locations within the Hunter region of NSW, seven of which only trapped *P. b. breviceps*, nine only trapped *P. norfolcensis,* while five found both species coexisting, and both species were live trapped (Figure 2). Trapping methodology and sampling sites were those of Knipler et al. [25]. Morphological data assisted with the assignment of individuals to either *P. b. breviceps* or *P. norfolcensis.* Measurements included head width, head length, body weight, right hind foot length, tail length, sex, reproductive status, and tail tip colour. During the trapping period, DNA was collected from Petaurus individuals using a sterilised 2 mm ear punch and the tissue stored in 80–95% ethanol at −20 °C.

### 2.2. Sequencing Mitochondrial DNA

Genomic DNA was extracted from 179 Petaurus ear tissue samples using the One-4-ALL Genomic Miniprep Kit (BioBasic Inc., Markham, ON, Canada) and DNeasy Blood and Tissue Kit (Qiagen, Hilden, Germany) as per the manufacturer’s instructions. If the identification to species remained uncertain despite morphological measurements, the mitochondrial (mtDNA) cytochrome b gene was sequenced using primer pair L14724 and H15149 following the instructions by Kocher et al. [26] and Irwin et al. [27]. This gene was chosen as it is the most sequenced mtDNA gene for Petaurus species, and therefore, there are numerous accessions available on GenBank in which to compare them [28]. PCR and post-PCR treatment methods were those of Knipler et al. [25]. Sequencing was undertaken using a 3130xl Genetic Analyzer (Applied Biosystems Pty Ltd., Scoresby, VIC, Australia). A total of 385 base pair sequences were edited and aligned with ChromasPro v 1.33 (Technelysium Pty Ltd., Brisbane, QLD, Australia) and BioEdit v 7.2.6.1 [29]. MtDNA cytochrome b sequences were then compared to *P. b. breviceps* and *P. norfolcensis* cytochrome b sequences through a BLASTN search of GenBank.

### 2.3. Next-Generation Sequencing

While morphology and mtDNA helped assign individuals to a species, they could not ascertain whether any individuals were *P. b. breviceps × P. norfolcensis* hybrids. At this point, the advent of next-generation sequencing and SNPs allowed important further analysis to occur. The remaining genomic DNA from the 179 glider individuals were sent to Diversity Arrays Technology, Canberra, Australia (DArTseq). DArTseq uses genome-complexity reduction methods to obtain thousands of genome-wide SNPs with restriction enzymes and next-generation sequencing technology. Additionally, DArTseq SNP data have been used to efficiently detect hybrids in frogs (*Litoria ewingii × L. paraewingi*) and clades of lizards (*Ctenophorus caudicinctus*) [8]. In our Petaurus study, DArTseq chose PstI-SphI restriction enzymes for optimal DNA fragmentation. Digestion and ligation followed that of Kilian et al. [30] with the addition of compatible adaptors for the PstI-SphI restriction enzymes. PCR fragments were amplified under the following conditions: denaturation of 94 °C for 1 min, followed by 30 cycles of denaturation (94 °C for 20 s), annealing (58 °C for 30 s) and extension (72 °C for 45 s) steps, and a final 7-min extension step at 72 °C. Amplified products were then equimolarly combined and subjected to a c-Bot bridge PCR and sequenced on an Illumina Hiseq2500 for 77 cycles.

DArTseq used their analytical pipelines to process the sequences and assign them to individual glider samples. Sequences were aligned to the Leadbeaters possum (*Gymnobelideus leadbeateri*) reference genome and markers were called with an average read depth of 20 reads per locus and a minimum sequence identity of 70%. After obtaining the data from DArTseq, the SNPs were filtered further using the *dartR* 1.1.11 package in R 4.0.2 [31,32]. Loci with >5% missing values were removed, loci with reproducibility <0.99 were removed, and those where hamming distance was <0.2. Any linked or monomorphic loci were also removed as well as those that deviated from the Hardy–Weinberg equilibrium. The remaining dataset was used for genomic analyses.

### 2.4. Statistical Analyses

We first ran a principal coordinates analysis (PCoA) in R 4.0.2 [32] to visualise the genetic distances and dissimilarities between individuals. A screeplot detected the number of PC axes that displayed most of the variation and these axes were plotted to examine the genetic differences between the two Petaurus species. Outlier individuals were flagged as potential hybrids. Pairwise F_ST_ calculated the genetic distances of species and potential hybrids using the *StAMPP* v1.6.1 package in R and 10,000 bootstraps. This package uses methods by Weir and Cockerham [33] and Wright [34]. Next, observed and expected heterozygosities were calculated for *P. b. breviceps, P. norfolcensis*, and potential hybrids with the *dartR* and *adegenet* 2.1.3 package in R [31,35]. Hybrids typically have high levels of heterozygosity (and reduced fitness) as their alleles come from different species [36], so these values were compared to see if there were high levels of heterozygosity in potential hybrid glider samples.

Both STRUCTURE and NewHybrids analyses have been used to identify hybrids in mammal research including *Felix catus × F. silvestris* [37,38], *Damaliscus pygargus pygargus × D. p. phillipsi* [39], and *Odocoileus hemionus hemionus × O. h. columbianus* [40]. STRUCTURE uses Bayesian clustering and Markov Chain Monte Carlo estimation to assign individuals to a group (predefined clusters, “K”) based on their allele frequencies [41]. Here, the SNP dataset was run through STRUCTURE v2.3.4 to assign glider individuals to genetic clusters (species) and identify potential hybrids based on admixture [42]. K values one to 15 were run eight times each with a 10,000-length burn-in period and 10,000 Monte Carlo Markov Chain replications. The optimal number of clusters (K) was then chosen using the Evanno method in STRUCTURE Harvester (Web v0.6.94) [43,44]. Individuals were considered pure to a cluster if the Q value was greater than 0.9 as per Wyk et al. [39], and subsequently considered admixed if 0.1 < Q < 0.9.

NewHybrids v1.0 uses a quantitative approach to identify hybrid individuals [45]. NewHybrids uses genotype frequencies to calculate the posterior probability that each Petaurus individual falls into the following categories: pure *P. norfolcensis*, pure *P. b. breviceps*, first generation hybrid (F1), second generation hybrid (F2), backcross to *P. b. breviceps* (0_Bx), and backcross to *P. norfolcensis* (1_Bx) (Table 1). The program only accepts 200 loci, so these were subsampled at random using the “gl2nhyb” function in the *dartR* package. A total of 10,000 sweeps were used for the burn-in and 10,000 sweeps were used for the Monte Carlo averages. Results for both STRUCTURE and NewHybrids were plotted in R.

## 3. Results

### 3.1. Morphological Features

Ninety-one individuals were identified as *P. b. breviceps* and 87 individuals were identified as *P. norfolcensis* based on morphological data. One individual (individual “GMP24”) was recorded as an adult male because of the large testes and active scent glands. It was initially believed to be *P. norfolcensis* because all previous captures at the site had been *P. norfolcensis* and its tail tip colour, head width, and head length measurements aligned with morphological features of the species (Figure 3). Despite this assumption, the sample was flagged for DNA species confirmation because the weight of the individual was 170 g and thus heavier than the average adult male *P. b. breviceps* (123.53 g) but lighter than the average adult *P. norfolcensis* (214.91 g) (Figure 3). No photographs were taken of the individual.

### 3.2. Mitochondrial DNA Sequence

As the species identity of individual GMP24 remained uncertain, its cytochrome b sequence was run through the BLASTN function on GenBank. The search returned a 100% identity match to a cytochrome b sequence of a *P. b. breviceps* sampled from Western Sydney, NSW, uploaded by Pavlova et al. [28] (Sequence ID FJ657662.1). The cytochrome b sequences of all other individuals matched the species that was identified through morphology.

### 3.3. Genome Wide SNPs: Testing for Hybrid

A total of 36,617 SNPs were obtained from DArTseq. After filtering, 10,111 SNPs remained from 179 Petaurus individuals and this dataset was used for the subsequent analyses. AMOVA results found that 76.340% of the genetic variation was apportioned between groups (*P. b. breviceps, P. norfolcensis*, and potential hybrid), 2.611% of the variation was apportioned between samples within groups, and 21.049% of the variation was apportioned between samples. The PCoA plot showed clear separation of the two Petaurus species on the first axis, with the PCoA Axis 1 accounting for a large proportion of the genetic variation (73.8%) and the PCoA Axis 2 accounting for only 1.1% of the genetic variation in the dataset (Figure 4). The PCoA Axis 2 showed greater variation for *P. norfolcensis* than for *P. b. breviceps.* When comparing the two species, the potential hybrid (“GMP24”) more closely resembled the nuclear DNA profile of *P. norfolcensis* (Figure 4), even though its mtDNA sequence matched that of *P. b. breviceps*. The pairwise genetic distances of species also conveyed this as *P. norfolcensis* had a smaller genetic distance to the potential hybrid than *P. b. breviceps* (F_ST_ = 0.346 vs. 0.671) (Table 2).

The average observed heterozygosity of *P. b. breviceps* and *P. norfolcensis* were similar to their expected heterozygosity values, however, the observed heterozygosity of the potential hybrid (GMP24) was higher than its expected heterozygosity as well as the observed heterozygosity of the two Petaurus species (potential hybrid Ho = 0.220, Hs = 0.114) (Table 3).

Both STRUCTURE and NewHybrids identified individual GMP24 as an admixed, hybrid individual. The STRUCTURE analysis detected the most likely number of clusters to be K = 2, indicative of the two glider species *P. b. breviceps* and *P. norfolcensis* (Figure 5). Ninety-one gliders were assigned pure to cluster K1 (*P. norfolcensis*), 87 gliders were assigned pure to cluster K2 (*P. b. breviceps*), and one individual (GMP24) was admixed with clusters K1 and K2 (Q_K1_ = 0.709, Q_K2_ = 0.291) (Figure 6). Similarly, NewHybrids assigned 91 gliders as pure *P. norfolcensis*, 87 gliders as pure *P. b. breviceps*, and one individual (GMP24) as a *P. norfolcensis* backcross (Table 1 and Figure 7). These assignments match the morphological assessments that were made in the field and suggest that individual GMP24 is a hybrid *P. norfolcensis* backcross. There was no evidence of F1 or F2 hybrids in the dataset.

## 4. Discussion

Prior to this study, there were no known cases of Petaurus hybridisation in the wild. Our study detected one hybrid individual in the Hunter region of NSW where the *P. b. breviceps* and *P. norfolcensis* distribution overlaps. The NewHybrids analysis detected a *P. norfolcensis* backcross, a very significant and valuable discovery as it signifies that a *P. b. breviceps × P. norfolcensis* F1 hybrid was fertile and able to reproduce with a *P. norfolcensis.* Additionally, we determined that the initial interspecies pairing was a female *P. b. breviceps* and a male *P. norfolcensis* due to the sequence of the maternally inherited cytochrome b gene. This F1 individual was determined to have been female in order to have passed on the *P. b. breviceps* mtDNA gene sequence to its offspring; the *P. norfolcensis* backcross. This exact combination has been reported in captivity by Fleay [2], who reported that a female *P. b. breviceps* from Victoria and a male *P. norfolcensis* from Queensland bred in captivity and produced a female offspring. Their offspring was fertile and reproduced with its *P. norfolcensis* father, which is supported by the findings of our study. This is significant as it means that hybrids are contributing to the gene pool of both *P. b. breviceps* and *P. norfolcensis* species in the location where it was trapped.

Our study detected backcrosses but did not detect F1 hybrids, resembling Canid research by Dufresnes et al. [46] and Macropus research by Neaves et al. [47]. Neaves et al. [47] investigated the marsupial hybridisation of *Macropus fuliginosus* and *Macropus giganteus.* They found no evidence of F1 hybrids, but a significant percentage of backcrosses. The findings by Neaves et al. [47] suggest that it is more common to encounter a backcross hybrid than an F1 hybrid in the wild, as it only takes one rare F1 hybrid to produce multiple backcrosses in their lifetime [47,48].

The location where the Petaurus hybrid was trapped is surrounded by urban development and heavily used roads, with the average daily traffic count in 2018 being 42,262 vehicles [49]. During our research, only *P. norfolcensis* were caught at this location during the trapping period (*n* = 3). Yadav et al. [50] discuss possible reasons for species hybridisation in the wild including small population size and habitat fragmentation. It is possible that both Petaurus species that occur at this site are in such low densities or sex ratios that they are forced to interact and mate with each other. Urbanisation and habitat fragmentation at this location is likely making it difficult for the species to disperse [50,51]. Increasing habitat connectivity in this area is essential.

While this study demonstrates evidence of fertile hybrids, it is important to note that there is currently no evidence of a female *P. norfolcensis* successfully producing a hybrid with a male *P. b. breviceps,* and there is no evidence of any pairings producing a fertile male F1 hybrid in captivity or in the wild. Haldane’s rule states that “when in the offspring of two different animal races one sex is absent, rare, or sterile, that sex is the heterozygous sex” [52]. Schilthuizen et al. [53] found 39 mammal species that obeyed this rule where male hybrids displayed sterility (*n* = 34) or inviability (*n* = 5). One example by Borodin et al. [54] examined F1 hybrids of *Thrichomys pachyurus*, *T. apereoides apereoides*, and *T. apereoides laurentius*, and found that all F1 males were sterile while only some F1 females were sterile. It is possible that the Petaurus genus also obeys this rule, as there is currently only evidence to support fertile female hybrids. If this holds true, then it is possible that the male *P. norfolcensis* backcross (individual GMP24) is sterile like other male backcrosses [50] and does not contribute to the population where it was sampled.

Hybrid glider individuals were predicted to have high levels of heterozygosity due to outbreeding between two separate species, resulting in a reduction in fitness [36]. As expected, higher heterozygosity was observed in the hybrid Petaurus individual (GMP24) when compared to the purebred Petaurus species. This is comparable to research on *Podarcis muralis* [55], *Canis lupis* [56], and *Zonotrichia leucophrys* subspecies [57]. Despite this, only one hybrid was identified in our study, so our results do not suggest that hybridisation is having a negative effect on the genetic structure or diversity of the *P. b. breviceps* or *P. norfolcensis* populations in the Hunter region. It does, however, suggest that the location in question needs added connectivity to surrounding habitat fragments to negate effects of small population sizes and habitat patch isolation. Despite only detecting one backcross hybrid, the results of this study are extremely important as *P. b. breviceps* and *P. norfolcensis* co-occur along the east coast of Australia and breed simultaneously. We have shown that they have the capability to hybridise in the wild, and there may be other cases of hybridisation where they coexist in low densities.

## 5. Conclusions

In conclusion, our study is the first to detect hybridisation of Petaurus species in the wild. Analyses of the genome-wide SNPs proved to be the most reliable way to identify hybrids, however, it is equally important to report morphological measurements and mtDNA sequences to gain insight into the fertility status of the hybrid individuals. Hybridisation must be considered in small, isolated habitat patches where both Petaurus species occur in low densities, especially if male hybrids prove to be sterile and take resources from other fertile individuals.

## Figures and Tables

**Figure 1 genes-12-01327-f001:**
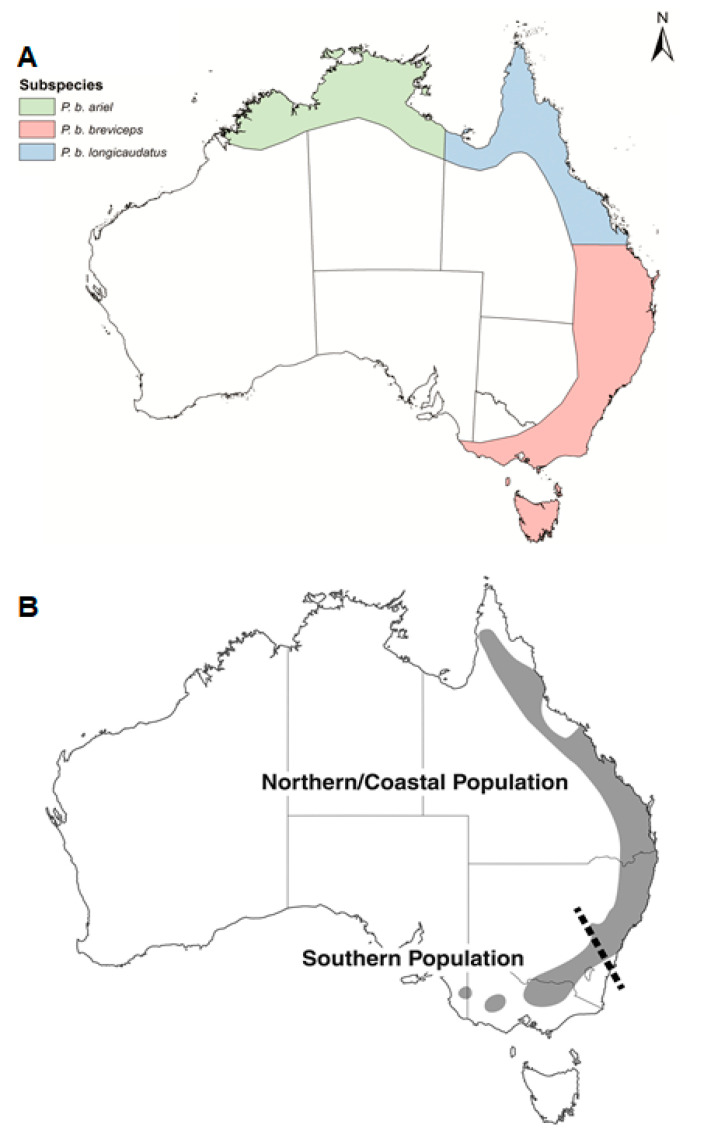
(**A**) *P. breviceps* subspecies distribution within Australia. Figure taken from Cremona et al. [15]; (**B**) *P. norfolcensis* distribution within Australia. Areas either side of the dashed line represent two genetically distinct populations. Figure taken from Crane et al. [16].

**Figure 2 genes-12-01327-f002:**
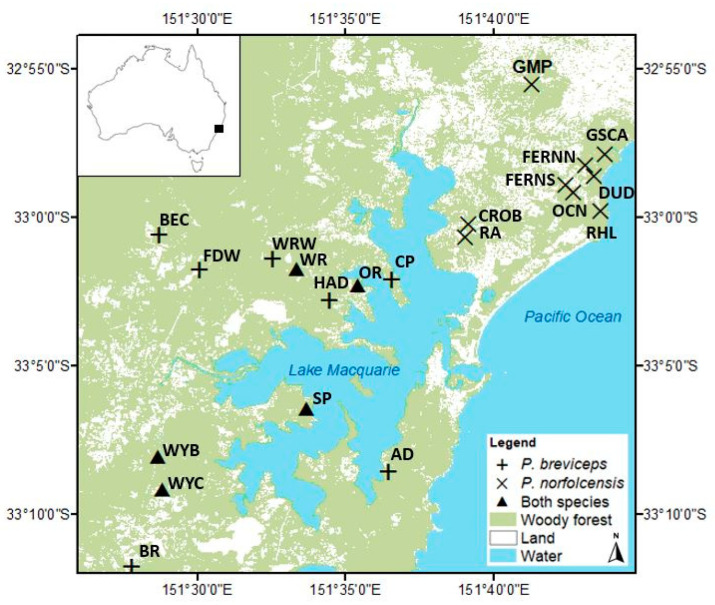
Twenty-one sampling locations of squirrel gliders (*P. norfolcensis*) and sugar gliders (*P. breviceps*) in the Lake Macquarie Local Government Area (NSW, Australia). The map shows locations where both species were present during the trapping period (triangles). The location of the potential hybrid is labelled “GMP”. *n* = 179.

**Figure 3 genes-12-01327-f003:**
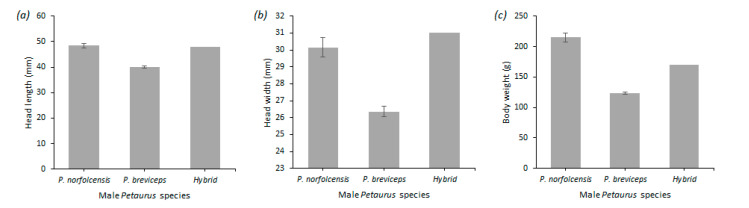
Mean morphological measurements of 23 male adult squirrel gliders (*P. norfolcensis*) and 47 male adult sugar gliders (*P. breviceps*) ± standard error. Morphological measurements of the hybrid (*P. breviceps* × *P. norfolcensis*) are also included. Measurements include (**a**) average head length of male Petaurus species in millimetres, (**b**) head width of male Petaurus species in millimetres, and (**c**) body weight of male Petaurus species in grams.

**Figure 4 genes-12-01327-f004:**
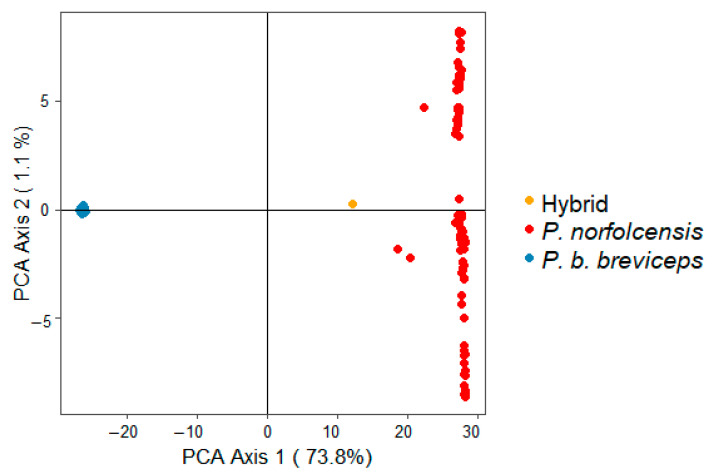
Principle coordinate analysis (PCoA) from genetic distances of 10,111 SNPs belonging to squirrel gliders (*P. norfolcensis*, *n* = 87) and sugar gliders (*P. b. breviceps*, *n* = 91). The potential hybrid individual (*P. b. breviceps* × *P. norfolcensis*) is coloured purple.

**Figure 5 genes-12-01327-f005:**
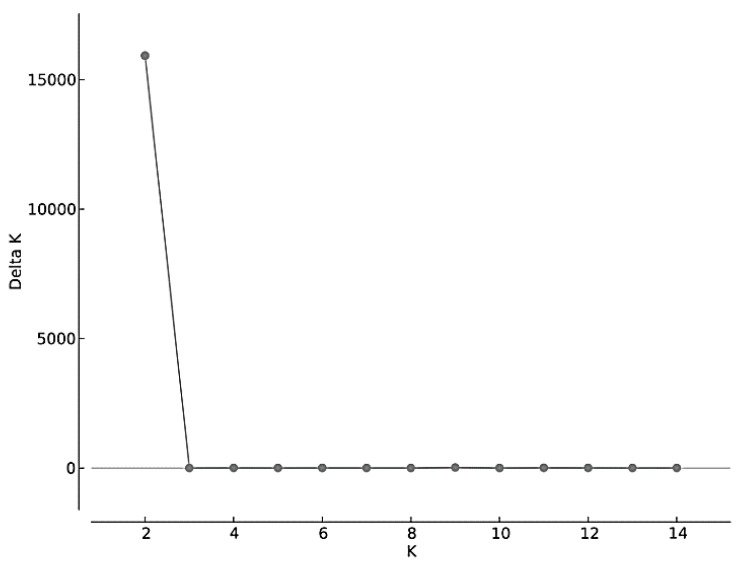
Delta K results from the Structure Harvester Evanno method. STRUCTURE tested 15 clusters (K = 1–15) with eight replicates each using 10,111 SNPs and 179 Petaurus individuals. Two clusters were detected (K = 2) using DeltaK = mean(|L”(K)|)/sd(L(K)).

**Figure 6 genes-12-01327-f006:**
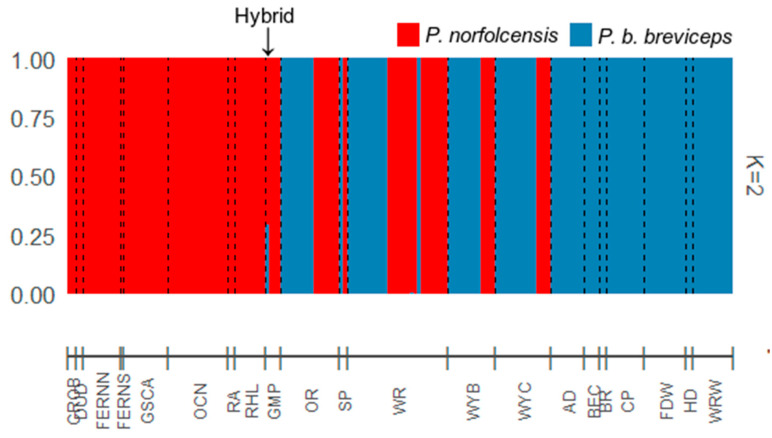
STRUCTURE admixture results using K = 2 and 10,111 SNPs. A total of 178 of the 179 Petaurus individuals were assigned “pure” to a cluster (*P. norfolcensis* or *P. b. breviceps* species), with one individual displaying admixture (hybrid *P. norfolcensis × P. b. breviceps*). Locations are listed along the bar on the *x* axis (refer to Figure 2 for more information).

**Figure 7 genes-12-01327-f007:**
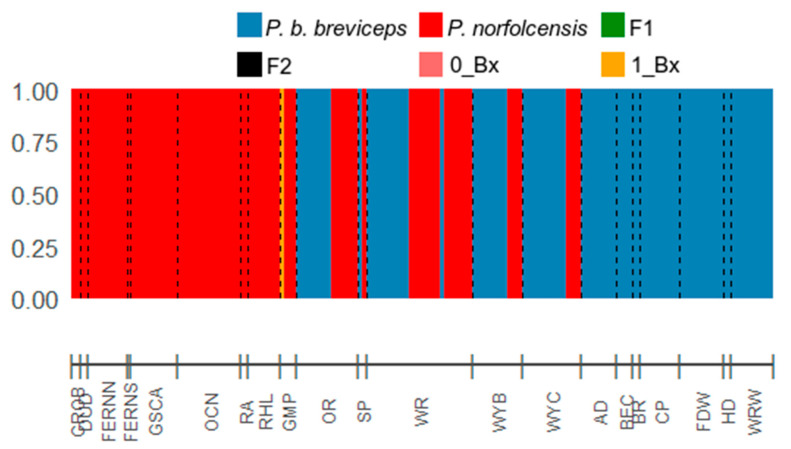
Results of NEWHYBRIDS analysis to identify hybrids between Petaurus species using 10,111 SNPs (*n* = 179). Possible results include pure squirrel glider (*P. norfolcensis*), pure sugar glider (*P. b. breviceps*), first generation hybrid (F1), second generation hybrid (F2), backcross to *P. b. breviceps* (0_Bx), and backcross to *P. norfolcensis* (1_Bx). Locations are listed along the bar on the *x* axis (refer to Figure 2 for more information). *y*-axis indicates assignment probability.

**Table 1 genes-12-01327-t001:** NewHybrids genotype frequency classes tested with 200 loci.

Genotype Frequency Classes	Product of	Expected Genotype Frequencies			
		*p* ^2^	pq	pq	*q* ^2^
Pure *P. b. breviceps*	2 × *P. b. breviceps*	1.00	0.00	0.00	0.00
Pure *P. norfolcensis*	2 × *P. norfolcensis*	0.00	0.00	0.00	1.00
First generation hybrid (F1)	*P. b. breviceps × P. norfolcensis*	0.00	0.50	0.50	0.00
Second generation hybrid (F2)	F1 × F1	0.25	0.25	0.25	0.25
*P. b. breviceps* backcross (0_Bx)	F1 × *P. b. breviceps*	0.50	0.25	0.25	0.00
*P. norfolcensis* backcross (1_Bx)	F1 × *P. norfolcensis*	0.00	0.25	0.25	0.50

**Table 2 genes-12-01327-t002:** Pairwise FST of *P. b. breviceps, P. norfolcensis*, and potential hybrid (individual GMP24) are shown below the diagonal, while significance is shown above the diagonal. * = significant (*p* < 0.05), NS = not significant (*p* > 0.05).

Species	*P. b. breviceps*	*P. norfolcensis*	Hybrid
*P. b. breviceps*	-	*	*
*P. norfolcensis*	0.767	-	*
Hybrid	0.671	0.346	-

**Table 3 genes-12-01327-t003:** Expected (Hs) and observed heterozygosity (Ho) of the two Petaurus species and the potential hybrid (individual GMP24).

Species	Hs	Ho
*P. b. breviceps*	0.084	0.082
*P. norfolcensis*	0.092	0.096
Hybrid	0.114	0.220

## Data Availability

Data are available from the authors upon request.

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
