# Peer review of "Genome-Wide SNPs Detect Hybridisation of Marsupial Gliders (Petaurus breviceps breviceps × Petaurus norfolcensis) in the Wild"

_genes, 2021, doi:10.3390/genes12091327_

Round 1

Reviewer 1 Report

The authors conducted a study investigating potential hybridisation between two species of gliders. They used morphological and genetic data. Among 179 individuals sampled, they found one potential hybrid. This is the first case of detection of hybridisation among the target species in the wild.

The text is well-written, materials and methods described in sufficient detail and the results adequately discussed.

I only have a few minor comments to address:

L19 and other places in the text: Remove the gap between the individual digits (e.g., 10,111, not 10, 111).

L42: Replace ‘is’ with ‘are’

Figure 1: What does the dashed line represent? Please add a description.

Figure 2: There should be an insert of a small map of Australia to provide a context of where the sampling locations are. Also, I suggest adding numbers or letters to each location so that they could be referred to within the text and other Figures.

L216-220: Did all of the other mtDNA sequences matched the species identified through morphology? Please report.

Figures 6 and 7: Following up on the comment about Figure 2, please add location numbers or letters to identify where the species were sampled from.

Author Response

Point 1: L19 and other places in the text: Remove the gap between the individual digits (e.g., 10,111, not 10, 111).

Response 1: These changes have been made.

Point 2: L42: Replace ‘is’ with ‘are’

Response 2: This change has been made.

Point 3: Figure 1: What does the dashed line represent? Please add a description.

Response 3: A description has been added to the Figure caption.

Point 4: Figure 2: There should be an insert of a small map of Australia to provide a context of where the sampling locations are. Also, I suggest adding numbers or letters to each location so that they could be referred to within the text and other Figures.

Response 4: These changes have been made.

Point 5: L216-220: Did all of the other mtDNA sequences matched the species identified through morphology? Please report.

Response 5: This has now been added to section 3.2.

Point 6: Figures 6 and 7: Following up on the comment about Figure 2, please add location numbers or letters to identify where the species were sampled from.

Response 6: Changes have been made to both Figures (location letters have been added).

Reviewer 2 Report

Dear authors,

The manuscript analyses the possible presence of P. breviceps and P. norfolcensis hybrids in Australia by genome-wide SNPs. The manuscript is well structured and written, but some minor changes are required. First, the authors must differentiate subsections within the Material and Methods section, including sampling, genetic analysis, and statistical analysis, among others. And secondly, the authors should expand the discussion with previous publications in other species, and above all, the authors should indicate the relevance of the results found.

Author Response

Point 1: First, the authors must differentiate subsections within the Material and Methods section, including sampling, genetic analysis, and statistical analysis, among others.

Response 1: These changes have been made.

Point 2: And secondly, the authors should expand the discussion with previous publications in other species

Response 2: The discussion has been expanded to include comparisons/mentions of other species.

Point 3: And above all, the authors should indicate the relevance of the results found.

Response 3: We have now addressed this in the discussion section.